The challenge of managing the commercial harvesting of the sea urchin Paracentrotus lividus: advanced approaches are required

http://orcid.org/0000-0003-0169-8044 Farina Simone 1 6 simone.farina@szn.it
Baroli Maura 1
Brundu Roberto 2
Conforti Alessandro 3
http://orcid.org/0000-0002-4469-2286 Cucco Andrea 3
De Falco Giovanni 3
Guala Ivan 1
http://orcid.org/0000-0001-9961-910X Guerzoni Stefano 1
Massaro Giorgio 3
http://orcid.org/0000-0002-0020-1780 Quattrocchi Giovanni 3
http://orcid.org/0000-0002-2208-3017 Romagnoni Giovanni 4 5
http://orcid.org/0000-0002-8521-8732 Brambilla Walter 3
1 IMC-International Marine Centre , Oristano , Italy
2 Marine Protected Area “Penisola del Sinis-Isola di Mal di Ventre” , Cabras, Oristano , Italy
3 CNR—IAS, National Research Council, Institute for the study of Anthropic impacts and Sustainability in the marine environment , Oristano , Italy
4 COISPA Tecnologia & Ricerca , Bari , Italy
5 Deptartment of Biosciences, University of Oslo, Centre for Ecological and Evolutionary Synthesis (CEES) , Oslo , Norway
6 Current Affiliation: Stazione Zoologica Anton Dohrn, Deptartment of Integrative Marine Ecology, Ischia Marine Centre , Ischia, Naples , Italy
Toonen Robert
Electronic publication date: 2020 Oct 8
Publication date: 2020
Volume: 8
Electronic Location ID: e10093
Received 2020 Apr 2; Accepted 2020 Sep 13
Copyright: © 2020 Farina et al.
Copyright year: 2020
Copyright holder: Farina et al.
License: This is an open access article distributed under the terms of the Creative Commons Attribution License, which permits unrestricted use, distribution, reproduction and adaptation in any medium and for any purpose provided that it is properly attributed. For attribution, the original author(s), title, publication source (PeerJ) and either DOI or URL of the article must be cited.
License URL: https://creativecommons.org/licenses/by/4.0/

Keywords: Sea urchins, Paracentrotus lividus, Population dynamic, Local fisheries management, Sustainable harvesting, Spatial management, Marine protected area, Environmental constrains, Stock sustainability, Marine coastal ecosystems

Funding: Interreg V A Italy France Maritime 2014–2020 Cooperation Program Italian Ministry of Research This work was supported financially by the Interreg V A Italy France Maritime 2014–2020 Cooperation Program, project “Gestione Integrata delle Reti ecologiche attraverso i Parchi e le Aree Marine—GIREPAM” (Asse 2–Lotto 3–PI 6C-OS 1) and RITMARE project (Subproject SP4, Work-Package 1, Actions 1, 2) funded by Italian Ministry of Research. The funders had no role in study design, data collection and analysis, decision to publish, or preparation of the manuscript.

==============================
Sea urchins act as a keystone herbivore in marine coastal ecosystems, regulating macrophyte density, which offers refuge for multiple species. In the Mediterranean Sea, both the sea urchin Paracentrotus lividus and fish preying on it are highly valuable target species for artisanal fisheries. As a consequence of the interactions between fish, sea urchins and macrophyte, fishing leads to trophic disorders with detrimental consequences for biodiversity and fisheries. In Sardinia (Western Mediterranean Sea), regulations for sea urchin harvesting have been in place since the mid 90s. However, given the important ecological role of P. lividus, the single-species fishery management may fail to take into account important ecosystem interactions. Hence, a deeper understanding of population dynamics, their dependance on environmental constraints and multispecies interactions may help to achieve long-term sustainable use of this resource. This work aims to highlight how sea urchin population structure varies spatially in relation to local environmental constraints and species interactions, with implications for their management. The study area (Sinis Peninsula, West Sardinia, Italy) that includes a Marine Reserve was divided into five sectors. These display combinations of the environmental constraints influencing sea urchin population dynamics, namely type of habitat (calcareous rock, granite, basalt, patchy and continuous meadows of Posidonia oceanica), average bottom current speed and predatory fish abundance. Size-frequency distribution of sea urchins under commercial size (<5 cm diameter size) assessed during the period from 2004 to 2007, before the population collapse in 2010, were compared for sectors and types of habitat. Specific correlations between recruits (0–1 cm diameter size) and bottom current speeds and between middle-sized sea urchins (2–5 cm diameter size) and predatory fish abundance were assessed. Parameters representing habitat spatial configuration (patch density, perimeter-to-area ratio, mean patch size, largest patch index, interspersion/juxtaposition index) were calculated and their influence on sea urchin density assessed. The density of sea urchins under commercial size was significantly higher in calcareous rock and was positively and significantly influenced by the density and average size of the rocky habitat patches. Recruits were significantly abundant in rocky habitats, while they were almost absent in P. oceanica meadows. The density of middle-sized sea urchins was more abundant in calcareous rock than in basalt, granite or P. oceanica. High densities of recruits resulted significantly correlated to low values of average bottom current speed, while a negative trend between the abundance of middle-sized sea urchins and predatory fish was found. Our results point out the need to account for the environmental constraints influencing local sea urchin density in fisheries management.

Introduction

The continuous decline of fishery catches during the last decades has pushed many fishermen to switch to new species at lower trophic levels (Anderson et al., 2011). One of the clearest examples from coastal ecosystems is the overexploitation of species involved in the typical tri-trophic interaction ‘‘fish-sea urchins-macroalgae’’ (Jackson et al., 2001). In the Mediterranean Sea, the interaction between the sea breams Diplodus spp. and Spaurus aurata, the sea urchin Paracentrotus lividus and coastal macroalgal forests follows this paradigm.

Paracentrotus lividus is one of the most important herbivores of Mediterranean benthic ecosystems (Hereu et al., 2005; Prado et al., 2012). The impact of overfishing through the impairment of predatory control on P. lividus determines a significant loss of macroalgal communities and biodiversity (Micheli et al., 2005; Giakoumi et al., 2012; Sala et al., 2012; Wallner-Hahn et al., 2015). For this reason, in the Mediterranean Sea it is widely accepted that P. lividus harvesting may be a potentially effective method for mitigating overgrazing in areas of severe overfishing (e.g. Piazzi & Ceccherelli, 2019).

Concurrently, targeted harvesting of sea urchin is progressively increasing worldwide (Andrew et al., 2002; James et al., 2016). Direct extraction is resulting in the collapse of local populations (Tegner & Dayton, 1977; Pennington, 1985; Levitan, Sewell & Chia, 1992; Levitan & Sewell, 1998) and in community-level effects, with rapid development of large, brown algae and changes in the composition of fish and benthic communities (Steneck et al., 2002). Sea urchin fisheries generally follow the short-term “boom-and-bust” pattern of many invertebrate fisheries. They start as a small-scale activity that undergoes a phase of rapid expansion followed by a phase of full exploitation that lead to the exhaustion of the resource (Andrew et al., 2002).

The key role played by sea urchins in benthic trophic interactions and in regulating subtidal communities needs to be considered in the development of sustainable urchin fisheries (Tegner & Dayton, 2000; Norderhaug et al., 2020).

In light of these considerations, an integrated management strategy for social and ecological systems has been developed in many regions where this situation has occurred (Moreno et al., 2006; Perry, Zhang & Harbo, 2002). There are a number of well-managed and sustainable sea urchin fisheries around the world that tend to rely on a good general knowledge of the biology of the urchin species present in the area as well as a sound understanding of the dynamics of sea urchin populations (James et al., 2016).

In New Zealand, for example, between 2002 and 2003, the sea urchin species Evechinus chloroticus was introduced into the quota management system of fishing, thanks to the support of the detailed biological information (Miller & Abraham, 2011). The quota management system is used to set the total allowed catch in twelve different fishing sectors according to the assessment of a set of biological criteria (Miller & Abraham, 2011). Fishing areas are classified in relation to growth conditions of the gonads, spawning rate, larval diffusion and the connectivity of local populations (Kritzer & Sale, 2004; James, Heath & Unwin, 2007; James & Heath, 2008; James et al., 2009; Wing, 2009).

In Mediterranean Sea, P. lividus is locally harvested in few regions by recreational and artisanal fisheries. In Sardinia (Italy, Western Mediterranean Sea) for example, where sea urchin populations have suffered unsustainable pressure since the early 2000s (Guidetti, Terlizzi & Boero, 2004; Pais et al., 2007, 2012; Ceccherelli et al., 2011), sea urchin harvesting is managed by a regional decree (Department of Environmental Protection Decree No. 276 of 3 March 1994 and subsequent amendments). This introduced a combination of measures including licenses, quotas, fishing techniques allowed, minimum size and seasonal closures. Regulation changed substantially after 2009: before 2009, professional fishermen (115–161 licenses) were authorized to collect up to 3,000 sea urchins per day by scuba diving along the entire coast of the Island (between November and April), with more restrictive regulations inside marine protected areas. After 2009, the number of regional licenses increased to 189, but stricter regulations have been introduced for the harvesting, transportation, storage and processing of the sea urchins (RAS, Autonomous Region of Sardinia, decree no. 2524/DecA/102 of 7 October 2009). Daily catches per fisherman were reduced to 2000, while the minimum catch size remained unchanged over the years (>5 cm diameter size).

The Peninsula of Sinis, in the central western coast of Sardinia, including the local Marine Protected Area “Penisola del Sinis, Isola di Mal di Ventre” (Marine Reserve from now on), is one of the main harvesting hotspots. Inside the Marine Reserve, some harvesting restrictions are applied: only resident professional fishermen are allowed to harvest sea urchins, for a maximum catch quota of 500 sea urchins per day per fisherman. The number of licensed fishermen varied from 125 in 2001 up to over 270 between 2004 and 2007 (including non-specialised artisanal fishers with special permission to harvest also sea urchins in this area). Nowadays, licenses issued decreased progressively down to 54 and recreational fishing has been banned (before 2009 it was allowed for residents only, with max 50 urchins per day).

Despite the tighter regulations in place since 2009, individuals larger than 5 cm diameter (minimum commercial size) are still infrequent both inside and outside the Marine Reserve. In fact, scientific monitoring in the Marine Reserve has shown a dramatic depletion both in commercial sizes (>5 cm diameter size) and in the whole population: sea urchin >5 cm firstly declined between 2004 and 2005, while the whole population dramatically declined since 2010 (Pieraccini, Coppa & De Lucia, 2016) with 65% and 75% reductions, respectively (Coppa et al., 2018). The year 2010 can be thus considered the onset of the crisis of P. lividus in the area.

Monitoring of sea urchin population structure (defined hereafter as abundance and age/size structure in the population) is performed on an ad-hoc basis in Sardinia in order to provide a scientific ground for stock assessment (Cau et al., 2007). The first regional surveys of sea urchins in Sardinia were carried out in 2001, 2003 and again in 2007. As one of the main harvesting hotspots, the Peninsula of Sinis has been closely monitored since 2004.

We used data of sea urchin density, by size class, collected between 2004 (first sampling) and 2007 (before the population collapse) to provide relevant information on population structure (under the commercial size of 5 cm diameter) when the impact of harvesting was still limited to the commercial class (Pieraccini, Coppa & De Lucia, 2016). These pre-collapse data represent a precious reference for understanding natural relationships between local population dynamics and the environmental constraints in the study area.

In general, recruitment and predation are considered the main ecological processes driving P. lividus population dynamics and shaping population structure locally (Fig. 1; Sala & Zabala, 1996; Goñi Beltrán de Garizurieta et al., 2000). Larval supply is strongly influenced by coastal hydrodynamics (Fenaux, Cellario & Rassoulzadegan, 1988; Goñi Beltrán de Garizurieta et al., 2000; Prado et al., 2012; Farina et al., 2018), while the nature of the substrate, the type of habitat and the abundance of predatory fish strongly influence settlement success and post-settlement survival (Boudouresque & Verlaque, 2001; Tomas, Turon & Romero, 2004; Hereu et al., 2005; Oliva et al., 2016). Settlement on rocky habitats is generally higher than in seagrass Posidonia oceanica, where the abundance of cryptic predators determines a high mortality rate (Tomas, Turon & Romero, 2004).

Figure 1 Diagram describing sea urchin population dynamics.

Letters represent different life stages of populations: (A) commercial stock and main reproducers of sea urchin populations, (B) larval supply for populations, (C) settlement in suitable habitats, (D) interactions with habitat structure for food and shelter, (E) predator–prey interactions with local predator community, (F) fishing pressure both on fish and sea urchins.

Once in the benthos, predation becomes the most important ecological driver of sea urchin distribution (Hereu et al., 2005; Tomas, Romero & Turon, 2005). The labrid Coris julis and the commercial sea breams Diplodus spp. and Sparus aurata represent the main predators of recruits and middle-sized sea urchins respectively (Sala, 1997; Guidetti, Boero & Bussotti, 2005). The predation risk of sea urchins strongly depends on the availability of shelters provided by the structure of the habitats and their spatial configuration (Farina et al., 2009, 2017; Pagès et al., 2012) until the urchins reach the safety size of ~5 cm (Guidetti et al., 2004).

We divided the study area into five fishing sectors with varying levels of environmental constraints influencing P. lividus recruitment and predation. Specifically, we focused on types of habitat (i.e., Calcareous rock substrate, Granite substrate, Basalt substrate, patchy and continuous meadows of Posidonia oceanica), and on a pool of variables describing habitat spatial configuration (patch density, perimeter-to-area ratio, mean patch size, largest patch index and the interspersion/juxtaposition index) which strongly influence shelter and food availability for sea urchins (Hereu et al., 2005; Farina et al., 2017). A circulation model of bottom current speed was used to approximate coastal hydrodynamics that strongly influence larval diffusion and sea urchin settlement (Farina et al., 2018). Finally, predatory fish abundance provides approximative information about potential predation activity along the fishing sectors (Guidetti, 2007).

Differences in the density of sea urchins under commercial size (<5 cm diameter size), recruit density (0–1 cm diameter size) and middle-sized sea urchin density (2–5 cm diameter size) are estimated for each fishing sector and type of habitat. Density of commercial size class (>5 cm diameter test) was also reported but not considered in the analysis because already compromised by harvesting in the study period (Pieraccini, Coppa & De Lucia, 2016).

Due to the absence of a direct estimation of predation and recruitment rates in these years, the importance of local hydrodynamics on population recovery and of predator activity on population structure are evaluated as a relationship (a) of the average bottom current speed to the density of recruits and (b) between the densities of predatory fish and middle-sized sea urchins, as the size-class range potentially vulnerable to fish predators (Sala & Zabala, 1996). Finally, the influence of spatial configurations of rocky habitats on the total density of sea urchins under commercial size (<5 cm diameter size) is estimated.

Taking advantage of the valuable dataset before the collapse of 2010, we attempt to capture the natural relationships between local population dynamics and the environmental constrains in the study area, providing a precious reference point to understand the mechanisms driving population structure before its collapse. This is a key step toward an improved comprehension of local population dynamics and a prerequisite for basing fishing quota allocation

In order to avoid a collapse of sea urchin populations in this area, and to achieve long term sustainability of the fishery, a major change in the management strategy is necessary (Ouréns, Naya & Freire, 2015). The aim of this study is therefore to provide evidence on the importance of embedding spatial and temporal environmental processes in the assessment of the stock sustainability towards a scientifically sound, ecosystem-based fisheries management that allows an integrated management of sea urchins and their predators in the Peninsula of Sinis.

Materials and Methods

Study area

The study area encompasses 40 km of the West Coast of Sardinia (Italy) between the Gulf of Oristano and Su Pallosu Bay (Peninsula of Sinis) (Fig. 2). This area includes the local Marine Protected Area of “Penisola del Sinis, Isola di Mal di Ventre”, which was established in 1997 and covers a surface of 250 Km2 (Fig. 2). The full protection area is 5 Km2 (Planes et al., 2008) while the remaining zones are intensively frequented by fishermen (Pieraccini, Coppa & De Lucia, 2016). The study area is limited to the bathymetry of 5 ± 1 m (mean depth at which the harvesters usually work) and it is subdivided into five fishing sectors (Table 1). Study sector 1 is identified in the portion of coast located outside the Marine Reserve from Su Pallosu Bay to the northern boundary of the Marine Reserve, including Cape Mannu (Fig. 2). Sectors 2 and 3 encompass the stretch of coast inside the Marine Reserve that is, exposed to the open sea, while sector 4 represents the Marine Reserve islands of Mal di Ventre and Catalano. Finally, sector 5, at the southern border of the Marine Reserve, includes part of the Gulf of Oristano.

Figure 2 Detailed digital mapping of geomorphology in the study area.

Colours indicate different sectors and types of habitats: Calcareous rock (CR in yellow ochre), Granite (GR in light red), Basalt (BA in red), Posidonia oceanica patchy meadow (PM in dark green), Posidonia oceanica continuous meadow (CM in light green) and sandy bottom (in yellow).

Table 1 Differences in average bottom current speed, predatory fish density, total sea urchin density and stock proportion between sectors with different rocky substrates and Posidonia oceanica meadows (in sector 5 only one observation was carried out).

Sector	Total area (Km2)	Average current speed (m/s)	Average predatory fish (ind/125 m2)	Total sea urchin density	Stock proportion (%)	
1	12.7	0.05 ± 0.003	69.6 ± 20	9.9 ± 1.1	15.1 ± 2.3	
2	5.1	0.09 ± 0.004	53.5 ± 7.2	8.9 ± 1.3	23.3 ± 2.2	
3	4.3	0.07 ± 0.007	74.8 ± 22.8	6.9 ± 1.5	28.7 ± 4.7	
4	3.8	0.10 ± 0.004	84.6 ± 12.6	7.5 ± 1.7	27.8 ± 3.6	
5	14.4	0.07 ± 0.003	–	2.5 ± 0.2	20.0 ± 1.8	

The seabed of the study area is composed of bedrock of different natures: Paleozoic granite basement, cropping out around Mal di Ventre Island; Pliocene basalt rock in the Cape San Marco area and surrounding Catalano Island (Fais, Klingele & Lecca, 1996; De Falco et al., 2003; Duncan et al., 2011; Conforti et al., 2016); and the Miocene and Quaternary Calcareous rocks located all along the study area coastline (Lecca & Carboni, 2007). These different types of substrate morphology influence the distribution of Posidonia oceanica; the meadow shows a patchy pattern where the matte is on the bedrock and a continuous pattern where the matte lies on the unconsolidated sediments (Fig. 2). The meadow is continuous on the eastern side of Mal di Ventre Island and inside the Gulf of Oristano, while Posidonia oceanica shows a patchy meadow pattern in the rest of the study area (De Falco et al., 2008).

Along the coastal area, the average bottom current speed (Fig. 3) strongly influences the abundance of sea urchin recruits (Farina et al., 2018). The water circulation in this area is mainly promoted by the action of the winds which are predominantly from the North-West, the Mistral wind, and from the South-West, the Libeccio wind, with average speeds of 7 m/s and with peaks around 20 m/s (Zecchetto et al., 2016). Such two prevalent wind regimes may generate intense flows towards the south, in the case of Mistral events, and weaker northward flows, in the case of Libeccio events. In both cases, within the Gulf of Oristano, recirculation cells develop in correspondence to the leeway side of the main two Gulf capes. We refer to Cucco et al. (2006, 2012) for a detailed description of sea current circulation in the study area.

Figure 3 Detailed digital mapping of hydrodynamism in the study area.

Map representing average bottom current speed obtained by the oceanographic model in the area of interest during six months from spawning time to the period of settlement (January–June).

Environmental constraints

Within the sectors, on the basis of the occurrence of different rocky substrates and Posidonia oceanica meadows, the environmental areas inhabited by sea urchins are defined as types of habitat (Abercrombie, Hickman & Johnson, 1966): Calcareous rock (CR), Granite (GR), Basalt (BA), Posidonia oceanica patchy meadow (PM) and Posidonia oceanica continuous meadow (CM).

The geomorphology was described through habitat mapping (Fig. 2). Available data consisted of morpho-bathymetric data, aerial images and several geo-datasets. To ease processing and data sharing among researchers, all available data were integrated and organized in a geodatabase implemented through a GIS and the software suite Geoinformation Enabling Toolkit StarterKit® (GET-IT), (Fugazza, Oggioni & Carrara, 2014; Pavesi et al., 2016; Lanucara et al., 2017; Brambilla et al., 2019) that was developed by researchers from the Italian National Research Council within the framework of the RITMARE research project.

The distribution and extent of habitats have been plotted to create a map with complete coverage of the seabed. Seafloor mapping has been made by imposing clear boundaries between different morphotypes (Fig. 2) to provide representations of how are they structured. Habitats alternate heterogeneously along the coast. A pool of variables describing the basic characteristic of their spatial configuration was estimated in each sector with the free software Fragstats 4.1 (McGarial & Marks, 1995). The estimated variables are Patch Density of types of habitat on the total landscape area (PD, patch/Km2), Perimeter-to-area ratio (P/A ratio, 1/m), Mean Patch Size of types of habitat (MPS, Km2), the Largest Patch Index (LPI, %) as the percentage of landscape area occupied by the largest patch of a type of habitat and Interspersion/Juxtaposition Index (IJI, %) which measures the degree of aggregation or “clumpiness” of a map based on adjacency of patches of the same type of habitat (O’Neill et al., 1988) (Table 2).

Table 2 Spatial configuration of sampled habitats for each study sector.

Sector	Habitat	Code	N° of samplings	Area (Km2)	PD (n/Km2)	P/A ratio (1/m)	MPS (Km2)	LPI (%)	IJI (%)	
1	Calcareous rock	CR-1	12	4.5	1.01	21.1	0.10	3	68.6	
Patchy meadow	PM-1	12	7.2	0.16	11.0	0.72	5.3	55.3	
Sand		–	1.0	–	–	–	–	–	
2	Calcareous rock	CR-2	15	2.5	0.62	15.2	0.31	17	98.9	
Patchy meadow	PM-2	7	2.3	0.08	13.2	2.28	17.8	46.3	
Sand		–	0.3	–	–	–	–	–	
3	Calcareous rock	CR-3	5	1.0	0.32	16.2	0.17	4.5	44.6	
Patchy meadow	PM-3	4	2.0	0.32	10.4	0.33	8.2	61.7	
Basalt	BA-3	4	0.1	0.05	0.3	0.18	0.6	33.9	
Sand		–	1.2	–	–	–	–	–	
4	Granite	GR-4	14	1.8	0.02	16.4	1.85	2.9	62.7	
Basalt	BA-4	3	0.1	0.02	21.4	0.08	0.1	0	
Patchy meadow		–	0.5	–	–	–	–	–	
Cont. meadow		–	1.4	–	–	–	–	–	
5	Cont. meadow	CM-5	3	11.1	2.6	2.1	3.7	42.6	5.5	
Other		–	0.1	–	–	–	–	–	
Sand		–	0.7	–	–	–	–	–	
Note:

A dash indicates that no samplings were carried out.

The average bottom current speed in the investigated area was obtained by means of a numerical modeling previously applicated in Farina et al. (2018). A three-dimensional hydrodynamic and wind wave model, SHYFEM–WWM (Umgiesser et al., 2004), previously used to reproduce the wind-wave and the 3D water circulation in the Western Sardinian Sea (Cucco et al., 2006, 2016; De Falco et al., 2008), was adopted.

In Farina et al. (2018), the authors reported the model solution for the biennium 2009 and 2010 since it is highly representative of the climate in the Sinis Peninsula (see Appendix 1 in there). The same solutions were used here to describe the water circulation in the first 10 m of water depth. Hourly data of the sea water speed at the bottom were averaged between January and June, corresponding to the period of active local recruitment (Table 1; Fig. 3) (Prado et al., 2012; Farina et al., 2018).

Finally, from a multi-year series of fish biomass data recollection, we extrapolated the abundance of sea urchin predatory fish for each sector from 2004 to 2007 with the exception of sector 5 (Marra et al., 2016). Data represent the abundance of the commercial sea breams Diplodus spp., Sparus aurata in the shallow water over the rocky bottoms (5 m in depth) collected using Underwater Visual Census (Table 1) (Marra et al., 2016). In these years, the reserve effect on fish biomass was not evident and no significant differences were detected between inside and outside the Marine Reserve with the exception of the sea bream that were more abundant inside (Marra et al., 2016; Table 1).

Sea urchin population structure

Sea urchin population structure was estimated for each type of habitat in the study sectors from a multi-year series of data from 2004 to 2007 (before the population collapsed). During this period, 79 samplings were carried out following a standard protocol at depths between 2 and 10 m (Planes et al., 2008). Data were collected as previously described in Farina et al. (2018). Specifically, for each site and type of habitat, sea urchin density was estimated as the number of individuals per square meter (ind/m2) and the sizes of the individuals (without spines) were measured with calipers to the closest mm.

For the statistical analysis, we define recruits as individuals with a diameter ≤1 cm that survived until approximately one year after their settlement (Ouréns et al., 2013) and middle-sized sea urchins as individuals of size class range 2–5 cm (diameter size), vulnerable to predatory fish. Recruits and middle-sized sea urchins together constitute the under commercial size. Sea urchins larger than 5 cm diameter represent the commercial stock and are also estimated but this size-class is not considered in the analysis carried out in relation with the environmental factors since its density was already reduced to low values by human activity before 2007 (Pieraccini, Coppa & De Lucia, 2016).

Sea urchin population density and structure are estimated for each type of habitat and sector for before their collapse (Table 3; Fig. 4). We carried out an analysis of variance of the sea urchin density for the under-commercial-sized (<5 cm diameter test), recruit-sized (0–1 cm diameter size) and middle-sized sea urchins (2–5 cm diameter test) function of “sector” and “habitat” as fixed factors. Assumptions of normal distribution and homogeneity of response variables were tested using D’Agostino-Pearson and Cochran’s tests. The densities of sea urchins under commercial size and those of the middle-size followed a normal distribution with unbalanced replicates and were analyzed with General Linear Model with Gaussian family distribution (Zuur et al., 2009). Whereas, given the non-normal distribution and the high amount of zeros in recruit density, the analysis of variance of recruits was performed with General Linear Model with Negative Binomial Distribution and certain zero Inflation in order to avoid biased parameter estimates and standard errors (Zuur et al., 2009). All the model validations are provided graphically (see Supplemental Material).

Table 3 Densities of sea urchin size-classes representing population structures.

Sector	Habitat	0–1 cm	1–2 cm	2–3 cm	3–4 cm	4–5 cm	5–6 cm (stock)	>6 cm (stock)	
1	Calcareous rock	3.6 ± 0.6	4.1 ± 0.7	2.9 ± 0.3	1.9 ± 0.2	2.4 ± 0.3	1.4 ± 0.2	0	
1	Patchy meadow	0	0	0.3 ± 0.1	1.1 ± 0.4	1.3 ± 0.3	0.7 ± 0.2	0.1 ± 0	
2	Calcareous rock	0.6 ± 0.2	1.2 ± 0.2	1.9 ± 0.3	1.8 ± 0.2	2.6 ± 0.4	2.4 ± 0.4	0.2 ± 0.1	
2	Patchy meadow	0	0.3 ± 0.1	1.2 ± 0.5	2 ± 0.4	2.1 ± 0.3	1.3 ± 0.3	0.3 ± 0.1	
3	Calcareous rock	0.9 ± 0.3	2.5 ± 0.6	2.3 ± 0.2	1.6 ± 0.3	1.5 ± 0.2	1.2 ± 0.3	0.1 ± 0	
3	Patchy meadow	0.1 ± 0.1	0.2 ± 0.1	0.6 ± 0.3	0.8 ± 0.4	1.2 ± 0.5	1 ± 0.2	0.1 ± 0.1	
3	Basalt	0	0.1 ± 0	0.1 ± 0	0.5 ± 0.2	2.5 ± 0.9	2.6 ± 1.1	0.6 ± 0.3	
4	Granite	1.9 ± 0.6	2.6 ± 0.4	1 ± 0.2	0.9 ± 0.2	1.4 ± 0.4	2.5 ± 0.5	0.8 ± 0.1	
4	Basalt	1.5 ± 1.1	1.2 ± 0.7	0.2 ± 0	0.1 ± 0.1	0.2 ± 0.1	0.5 ± 0.3	0.3 ± 0.2	
5	Continuous meadow	0	0	0.3 ± 0.3	0.5 ± 0.3	1.1 ± 0	0.6 ± 0.5	0	
Note:

Mean sea urchins densities of the size-class range representing population structure in the different types of habitat. Size-class ranges 0–1cm and 2–5cm diameter represent recruits and middle-sized sea urchins respectively.

Figure 4 Graphs representing different population structures.

Populations of each type of habitat in each sector: (A) calcareous rock of sector 1 (CR-1), (B) patchy meadow of sector 1 (PM-1), (C) calcareous rock of sector 2 (CR-2), (D) patchy meadow of sector 2 (PM-2), (E) calcareous rock of sector 3 (CR-3), (F) patchy meadow of sector 3 (PM-3), (G) basalt of sector 3 (BA-3), (H) granite of sector 4 (GR-4), (I) basalt of sector 4 (BA-4) and (J) continuous meadow of sector 5 (CM-5).

Relationship between population structure and environmental conditions

Spearman’s rank correlation coefficient as a non-parametric measure of rank correlation was carried out between non-normal distribution values of recruit density and the average bottom current speed, while Pearson’s rank correlation, as a parametric linear regression test, was used to estimate the statistical relationship between normally distributed values of density for middle-sized sea urchins and the density of predatory fish.

The Generalized Linear Model (GLM) with Poisson family distribution was performed in order to assess the influences of a pool of variables representing the basic configuration of the rocky habitats on total sea urchin density for those under the commercial size. Patch Density, Perimeter-to-area ratio, Mean Patch Size, Largest Patch Index and Interspersion/Juxtaposition Index (IJI, %) are previously estimated (see above) and used as predictors for sea urchins density (commercial stock excluded). The stepwise forward regression technique was used to select the more conservative model (Whittingham et al., 2006).

Given the lack of data on fish visual census in Posidonia oceanica and since recruits are considerably underestimated inside meadows (Oliva et al., 2016), the patchy meadow and continuous meadow types of habitats were excluded a priori from all these analyses. Analyses were performed using R Studio (R Core Team, 2014).

Results

Environmental constraints

Sector 1 is the largest sector with an area of 12.7 Km2 (Fig. 2). The average current speed was 0.05 ± 0.003 m/s (Fig. 3) which was the slowest current measured in the recruitment period (from January to June; see Table 1). Conversely, sector 4 is the smallest sector with a total area of 3.8 Km2 (Fig. 2) and the highest predatory fish abundance of 84.6 ± 12.6 ind/125 m2 (Table 1). Sectors 2 and 3 extend 5.1 Km2 and 4.4 Km2 respectively (Fig. 2) with intermediate values for both bottom current speed average and predatory fish abundance (see Table 1 and Fig. 3). Finally, sector 5 covers a total area of 11.9 Km2 with a similar bottom current speed average for sector 3, while data on predatory fish abundance were not available (Table 1).

In sector 1, Posidonia oceanica patchy meadow (PM-1) is the most extended habitat with a surface of 7.2 Km2, while Calcareous rock (CR-1) covers 4.5 km2 with a Patch Density of 1.0 per Km2 In sector 2, Calcareous rock (CR-2) presents the most extensive surface of the habitats in the sector with the highest patch aggregation (98.9% of IJI) of all the habitats in the study area (see Table 2).

In sector 3, Posidonia oceanica patchy meadow (PM-3) and Calcareous rock (CR-3) cover a surface of 2 and 1 Km2 respectively and both habitats present a Patch Density of 0.32 per Km2 (Table 2). Basalt (BA-3) is distributed over 0.1 km2 in both sectors 3 and 4. In Sector 4, Granite (GR-4) covers 1.8 Km2 (Table 2). Finally, in sector 5 Posidonia oceanica continuous meadow (CM-5) represents 11.1 Km2 of the surface in the sector with the largest patch covering 42.6% of the total area (Table 2). Continuous meadow is also present in sector 4 but sea urchins have never been sampled there. A variable proportion of sandy bottom is present in all the sectors with the exception of sector 4.

Sea urchin population structure

Sector 1, located outside the Marine Reserve, presented the highest sea urchin density of 9.9 ± 1.1 ind/m2, but the lowest proportion of commercial stock (15.1%; Table 1). Inside the Marine Reserve, sea urchin density ranged from the low density of sector 5 of 2.5 ± 0.2 ind/m2, with a proportion of 20% commercial stock, to the high density of 9.8 ± 1.2 ind/m2 in sector 4 and with a proportion of commercial stock of 28.7% (Table 1).

The density of specimens under commercial size differ significantly between habitats (p-value < 0.001) and between sectors (p-value = 0.02; Table 4A). Among the types of habitat, the highest sea urchin density for specimens under commercial size was found in CR-1: 16.3 ± 1.4 ind/m2 (Fig. 4; Table 3). High values were also found in CR-2, CR-3 (10.6 ± 1.3 and 10.1 ± 0.8 ind/m2 respectively) and in GR-4 (11 ± 1.1 ind/m2). Otherwise, the lowest sea urchin density was estimated in correspondence to CM-5 (Table 3).

Table 4 Analysis of deviance table GLM model.

	Factor	DF	DR	F-value	p-Value	
(A) Response variable						
Density of under-commercial size	Sector	4	193.29	3.2592	0.01638	
Habitat	3	801.33	18.0157	8.534e−09	
Residual	79				
(B) Response variable						
Density of recruit	Sector	3	44.68	6.038	0.00103	
Habitat	3	101.21	13.677	4.26e−07	
Residual	69	7.40			
(C) Response variable						
Density of middle-size	Sector	4	125.61	5.1458	0.0010676	
Habitat	3	130.73	7.1410	0.0002935	
Residual	78				
Note:

(A) Density of under-commercial-size, (B) density of recruit and (C) density of middle-size sea urchins in function of Sector and Habitat as fixed factors. DF, degrees of freedom; DR, deviance residual; F, F statistics; P, probability of Type I error.

Densities for both recruits and middle-sized sea urchins were significantly different among habitats and sectors (p-values < 0.001; Fig. 4 and Table 4B and 4C). Recruits were significantly more abundant in sector 1 and sector 4 where they were 3.6 ± 0.6 ind/m2 in CR-1, 1.9 ± 0.6 ind/m2 in GR-4 and 1.5 ± 1.1 ind/m2 in BA-4 (Tables 3 and 4B). Meanwhile, no recruits were found in PM-1, PM-2, BA-3 and CM-5 (Fig. 4; Table 3). The highest average value of density for middle-sized sea urchins was found in CR-1 at (7.2 ± 0.8 ind/m). Average density values for CR-2 and CR-3 (6.3 ± 0.9 ind/m2 and 5.4 ± 0.7 ind/m2 respectively) were higher than for BA-3, BA-4 and GR-4 (3.1 ± 1.1, 0.5 ± 0.2 ind/m2 and 3.3 ± 0.8 ind/m2 respectively) (Table 3). In CM-5, the density of middle-sized sea urchins was 1.9 ± 0.5 ind/m2 (Fig. 4; Table 3).

Relationship between population structure and environmental conditions

Values of recruit density in rocky habitats (Calcareous rocky, Basalt and Granite) follow a non-normal distribution due to the high number of sampled zeros. Accordingly, Speraman’s non-parametrical rank correlation test was performed between recruit density and average bottom current speed and a negative significant relationship was found (Spearman’s rank correlation p-value = 0.002932; rho = −0.3972998; Fig. 5A). The density of middle-sized sea urchins following normal distribution was correlated to the predatory fish density using Peason’s correlation test and the variables resulted in a significant negative correlation (Pearson’s correlation p-value = 0.04268, correlation coefficient = −0.5118654; Fig. 5B).

Figure 5 Graphs representing relationships between sea urchin densities and environmental constraints.

In rocky habitats (A) density of recruits is correlated with the average bottom current speed (Spearman’s rank correlation) and (B) density of middle-sized sea urchins with predatory fish density (Pearson’s correlation) Number of points used in the graph a corresponds to the sea urchin sampling stations while in the graph b to the stations of fish visual census.

The General Linear Model highlights high significant influences of Patch Density (PD; p-value < 0.001) and significant influence of the Mean Patch Size (MPS; p-value < 0.001) on sea urchin density for specimens under commercial size. The proportion of the variance explained by the Minimal Adequate Model is roughly 50% (see Table 5 and Fig. S5).

Table 5 Generalized Linear Model (GLM) showing the effects of the assessed explanatory variables on the density of commercial under-sized classes (TD < 5 cm).

Response variable	Effect	Estimate	SE	Z-value	p-Value	
Full model	
	MPS
PD
IJI
LPI
P/A ratio	0.310135
1.073995
0.001116
−0.011801
0.007291	0.120982
0.201901
0.004847
0.019027
0.015867	2.563
5.319
0.230
−0.620
0.460	0.0104
1.04e−07
0.8179
0.5351
0.6459	
Minimal adequate model	
Sea urchin density	MPS
PD	0.3758
1.1459	0.0861
0.1381	4.365
8.300	1.27e−05
2e−16	
Note:

The Minimal Adequate Model (AIC = 290.8; R-square = 0.468) was obtained starting from Full Model (AIC = 295.5; R-square = 0.476) through the stepwise forward regression technique (Anova p-value = 0.55). Coefficient estimates (Estimate), standard errors (SE), z-values and significance levels (p-value) for variables are provided for fixed effects. Significant effects are given in bold.

Discussion

The surveys carried out between 2004 and 2007 revealed conspicuous differences in sea urchin density across fishing sectors and types of habitat. In general, rocky habitats of Calcareous rock, Basalt and Granite supported larger sea urchin populations than the habitats characterized by Posidonia oceanica.

Excluding the commercial component of the stock whose density was distorted by intensive fishing, the sea urchin density for specimens under commercial size in rocky habitats was significantly higher in Calcareous rock. Moreover, considering results obtained from the analysis on the spatial configuration, larger and more dense patches seems to further enhance sea urchin density in Calcareous rock.

In fact sector 1 was outside the Marine Reserve and had the lowest proportion of commercial stock, Calcareous rock in this sector presented a large surface (4.5 Km2) with high Patch Density (one patch per Km2) which supported a density of sea urchin under commercial size approximately twice that of Calcareous rock in sectors 2 and 3. Specifically, recruit density in Calcareous rock of sector 1 was 6 and 4 times higher than in the Calcareous rock of sectors 2 and 3 respectively, and 2 and 2.5 times higher than in the Granite and Basalt of sector 4 respectively. The density of middle-sized sea urchins resulted more than 2 times higher in Calcareous rock in general (sectors 1, 2 and 3) than in the Granite and Basalt (sector 4). Finally, in the Posidonia oceanica patchy meadows, recruit density was negligible everywhere, and similarly for the middle—sized sea urchins in the continuous meadow of sector 5.

Population structures analyzed responded to the high variability of the environmental constraints observed along this stretch of coast. From January to June, when spawning occurs (Loi et al., 2017) and settlement is supposed to be over (estimating 20–30 days for the planktonic phase once the eggs are fertilized, Lozano et al., 1995), the average bottom current speed was slowest in sector 1. It was almost half the speed of sectors 2 and 4 and a third less than in sectors 3 and 5. The weak, negative correlation between recruits and bottom current speed is a distant approximation of the real influence of hydrodinamics on population recruitment. This correlation was performed due to the lack of data on larvae and settlers during these years. In general, the influence of current on recruitment can serve as an indicator of effective connectivity between areas (Romagnoni et al., 2020). However, sea urchin density during the post-settlement phase experiences important decreases due to predators (Hereu, Zabala & Sala, 2008) and, as consequence, bottom current speed should be more closely correlated to larvae and settlers than to recruits. The low values of the average bottom current speed (<0.1 m/s) correspond to recruit densities above 3.5 ind/m2. It is noteworthy that the average bottom current speed on the Calcareous rock of sector 1 is always below this critical threshold. Accordingly, this condition seems to support the existence of local standing circulation structures that determine a higher regime of natural recruitment (Farina et al., 2018).

After recruitment, predation is the second main process regulating sea urchin population structure on a local scale (Guidetti, 2004; Hereu, Zabala & Sala, 2008; Boada et al., 2015). Adult sea urchins are preyed on by few fish species, especially the sea breams which are targeted by artisanal fisheries (Guidetti, 2006). During 2004–2007, there was a negative correlation between the abundances of the sea breams and middle-sized sea urchins. Low abundances of predatory fish were found outside the Marine Reserve in sector 1, most likely due to the strong pressure exerted by recreational spearfishing (Marra et al., 2016). Conversely, in sector 4—the Islands inside the Marine Reserve—the density of sea breams was higher than in the other sectors (Marra et al., 2016).

The reduced accessibility of the islands compared to the other coastal sectors could have offered seabreams protection from recreational spear fishermen, allowing higher abundance in this sector. Consistently with this theory, the lowest density of middle-sized sea urchins was found in sector 4, supporting the possibility of a higher level of predation in this area due to higher density of predatory fish.

Moreover, predator activity is generally influenced by an increase in habitat edges (Bender, Contreras & Fahrig, 1998; Kondoh, 2003; Prado et al., 2008; Farina et al., 2017). This is typically caused by fragmentation processes, which generally result in increasing habitat complexity as patch perimeter-to- area ratios increase (Ranney, Bruner & Levenson, 1981). The opposite condition is designed by the calcareous rock in sector 1. High-density patches with large surfaces dampen visual predation of fish providing efficient shelters to middle-sized sea urchins and recruits as well (Hereu et al., 2005).

Our results suggest how environmental constraints exert an important influence on sea urchin population dynamics and population structures and are not quite as homogenous as it might seem along this stretch of coast. Such heterogeneity might indicate that the sea urchin population in this region could be composed of multiple, smaller populations with their own dynamics, potentially connected via larval dispersion. In fact, the long planktonic early life-stage (between 20 and 30 days according to Lozano et al., 1995) could theoretically make sea urchin populations well connected (López et al., 1998; Morgan et al., 2000; Prado et al., 2012; Treml et al., 2012). However, in this area larval dispersion could be strongly dependent on the bottom current speed (Farina et al., 2018). The presence of “source–sink” dynamics via larval dispersal mediated by bottom current speed could affect conservation and management strategies for sustainable fisheries (Romagnoni et al., 2020; Kritzer & Sale, 2004; Kerr et al., 2017). This is especially important for conservation requirements in a Marine Reserve (Paterno et al., 2017) and it is an aspect of concern for future research in this area.

The strength of connectivity depends strongly on the abundance of reproducers. Since commercial harvesting depletes the main reproducers (i.e., size class >5 cm), middle-sized sea urchins play an important role in the population’s recovery (Loi et al., 2017). However, this size class is highly vulnerable to predators (Sala & Zabala, 1996). For this reason, harvesting of this size class should be limited in conditions of high predation mortality. This could be the case of the Islands of sector 4, where the sea urchin population seemed to suffer a higher predation pressure than in the other sectors.

High proportions of middle-sized sea urchins in patchy meadows are found in accordance with the efficient shelter that Posidonia oceanica leaves provide from the visual mechanism of predatory fish (Farina et al., 2009). However, the three-dimensional structure of large seagrass meadows can become a “death trap” in the presence of high densities of bottom predators (Farina et al., 2014, 2016; Schmidt & Kuijper, 2015). For example, sector 5, in the Gulf of Oristano, is characterized by a large, continuous meadow of Posidonia oceanica (De Falco et al., 2008). Here, the low density of P. lividus could be related to the abundance of whelks (e.g., Hexalplex trunculus, authors personal observation), which is known as effective predators of sea urchins (Farina et al., 2016) and whose proliferation is probably favored by the bio-deposits that accumulate beneath the nearby mussel farms (Inglis & Gust, 2003).

In order to achieve long-term sustainable exploitation of marine resources, fisheries management should account for the key processes regulating population dynamics on a relevant spatial and temporal scale (Hilborn & Walters, 1992). Despite the approximations and data limitations, our analysis identified spatial heterogeneity in sea urchin stock abundance related to local conditions as well as emergent natural relationships between sea urchin population dynamics, their ecological drivers and the environmental constraints in this area. These findings may be of help in advancing management in the area.

In this system, the continuous and inexorable decrease of sea urchins since 2007 has been followed by frantic adjustments in management measures. Regular stock assessment for P. lividus has been proposed to provide a scientific basis for management in Sardinia based on ad-hoc data and on regular scientific monitoring of sea urchin density (Addis et al., 2009). However, given the key role played by P. lividus in coastal ecosystems, advanced approaches could be required to provide long-term sustainable exploitation that considers the importance of environmental constraints in influencing local sea urchin population structure through the ecological drivers of recruitment and predation (Miller & Abraham, 2011). For example, fisheries models with explicit inclusion of spatial dynamics (Kritzer & Sale, 2004; Kerr et al., 2017) or multispecies dynamics allow for the testing of alternative management strategies (Christensen & Walters, 2004; Fulton et al., 2004; Spedicato et al., 2010).

When informed with ecologically relevant dynamics, such as those identified in the current study, these tools could complement single-species stock assessment in identifying management measures for regulating the fishing effort on specific components of the populations (for example, reducing harvesting of key size classes) or areas (e.g., areas with high predation mortality, or in key “source” areas).

The Peninsula of Sinis offers a unique case study, where ecological and economic knowledge is building up, and the involved stakeholders, including small-scale fisheries organizations, Marine Reserve and local administration are demanding a science-based management system. There is currently momentum toward the implementation of a long-term vision which entails a data collection procedure aiming to provide management strategies for the sustainable management of sea urchin fisheries. These would combine the objectives of conservation of ecological features and of traditional and socio-economics values. Moreover, achieving sustainable fishing of both resources-sea urchins and sea breams-should enhance, as a cascade effect, the conservation prospective for macrophyte communities, which are pivotal for ensuring a high environmental quality and support nursery of other benthic species.

Supplemental Information

Supplemental Information 1 Supplemental material.

Click here for additional data file.

Supplemental Information 2 Dataset.

Click here for additional data file.

Supplemental Information 3 R codes of statistical analysis.

Click here for additional data file.

The authors want to thank all researchers and students who contributed to creating a historical dataset on sea urchin density in this area through their work.

Additional Information and Declarations

Competing Interests

Author Contributions

Data Availability

Andrea Cucco is an Academic Editor for PeerJ.

Simone Farina conceived and designed the experiments, performed the experiments, analyzed the data, prepared figures and/or tables, authored or reviewed drafts of the paper, and approved the final draft.

Maura Baroli performed the experiments, authored or reviewed drafts of the paper, and approved the final draft.

Roberto Brundu performed the experiments, authored or reviewed drafts of the paper, and approved the final draft.

Alessandro Conforti performed the experiments, authored or reviewed drafts of the paper, and approved the final draft.

Andrea Cucco performed the experiments, analyzed the data, prepared figures and/or tables, authored or reviewed drafts of the paper, and approved the final draft.

Giovanni De Falco performed the experiments, authored or reviewed drafts of the paper, and approved the final draft.

Ivan Guala performed the experiments, authored or reviewed drafts of the paper, and approved the final draft.

Stefano Guerzoni conceived and designed the experiments, performed the experiments, authored or reviewed drafts of the paper, and approved the final draft.

Giorgio Massaro performed the experiments, authored or reviewed drafts of the paper, and approved the final draft.

Giovanni Quattrocchi performed the experiments, analyzed the data, authored or reviewed drafts of the paper, and approved the final draft.

Giovanni Romagnoni conceived and designed the experiments, performed the experiments, authored or reviewed drafts of the paper, and approved the final draft.

Walter Brambilla conceived and designed the experiments, performed the experiments, analyzed the data, prepared figures and/or tables, authored or reviewed drafts of the paper, and approved the final draft.

The following information was supplied regarding data availability:

The dataset and R codes for statistical analysis are available in the Supplemental Files.

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
