# Peer review of "The challenge of managing the commercial harvesting of the sea urchin Paracentrotus lividus: advanced approaches are required"

_PeerJ, doi:10.7717/peerj.10093_

## Round 0.1 · original submission · Major Revisions

I now have reviews back from 2 expert referees, both of whom felt that the paper needed extensive revision before it could be considered for publication. In particular, the English usage needs to be improved for communication, and the main goals and hypotheses being tested in the study were unclear to the referees (perhaps due to the language issue). The manuscript could benefit from careful editing using an English Language Service or by a native-equivalent English speaker. In either case, these reviewers have taken the time to offer detailed constructive feedback on the areas for improvement of the manuscript. Questions raised by the referees regarding clarity in the experimental design and time frame of the data being analyzed, the interpretation of the statistics and effect sizes, and the conclusion being drawn from the study must be addressed in a revision for the paper to be given further consideration.

Reviewer 1 ·

Basic reporting

This manuscript presents an interesting dataset on field observations carried out in shallow subtidal rocky reefs of Western Sardinia (Italy) with the goal of analyzing patterns and processes in abundance and population structure of the edible sea urchin Paracentrotus lividus in relation to biological and physical properties of the studied areas before a major decline in local sea urchin population. Results are discussed in the context of the management of the sea urchin fishery, and how knowledge of key ecological processes at relevant scales should be integrated with urchin stock management and conservation strategies to promote sustainability.
In general, the manuscript is well written and comprehensible. Nevertheless, I found it hard to follow some of the contents of the text, some graphs and tables require further description and labelling, and the raw data and code should be curated before making it publicly available.
At present, I see authors need to accommodate/clarify some issues before the paper is accepted for publication (please see specific comments to the manuscript in the general comments section below). In my opinion there are several points in each section of the paper that require further clarification in order to improve the interpretation and readability of the manuscript. The authors should also consider having their manuscript proofread by a native speaker before resubmitting.

Experimental design

The main goals and hypotheses being tested in the present study are not clearly identified.
Methods described with sufficient detail, but some aspect could be improved (please see specific comments to the manuscript in the general comments section below)

Validity of the findings

The findings are interesting, certainly contribute to increase the knowledge regarding local sea urchin populations and may have direct application on resource management and conservation strategies. However, there is room for improvement of the discussion and conclusion sections (please see specific comments to the manuscript in the general comments section below).

Additional comments

Specific comments to the manuscript:

Title

Consider mentioning in the title that the target species in this study is a sea urchin, as I’m sure some PeerJ reader will not be familiar with the species Paracentrotus lividus.

Abstract

In my opinion, the abstract doesn’t mention important background for the pertinence of the study, like the importance of this commercial activity in the study region and the management strategies in place (MPA, harvesting zones, etc..) and the severe decline observed in sea urchin abundance from 2007 onwards that motived this study.

Consider mentioning the species Paracentrotus lividus is a sea urchin the first time it is mentioned. You only refer to sea urchins about halfway through the abstract.

Line 20: ”…. functional species Paracentrotus lividus…”. Please describe the functional species concept in this context. It is not clear to me what authors are aiming.

Line 22-26: “Sea urchin density and …. “. Consider rephrasing this sentence where a brief description of the methodology is intended. There is no mention to which environmental constrains were addressed, which types of habitats were studied and what is meant by spatial organization in the context of the study. Also, the “inside/outside the local MPA” issue, which refers to the different sectors assessed in this study, lacks some contextualization in the abstract. Authors do not mention when the observations were carried out.

Introduction

Line 41: “…and reducing…”. Change to “reduce”.

Line 48: “…”boom-and-bust” patter…” . Missing an “n” in pattern.

Line 52: “Thus… ”. Consider rephrasing. “Thus” does not read well.

Line 59-62: Please develop this paragraph, briefly describing the main management strategies developed in the regions that are mentioned.

Line 67-69: “… seems to be much more important than…”. Please further describe what you mean by “more important”, in what sense?

Line 72: “… since decades”. Since when? Please provide a more specific time-frame.

Line 72-76. Authors should describe in further detail the harvesting/conservation regulation in the Peninsula de Sinis region in time and space. This should be explicit in the introduction section as it is important for the contextualization of the study. Later they provide more info on this issue in the methods section, but in my opinion, this should be included in the intro section.

Line 77: “Despite these restrictions… “. What restrictions? Please refer to the previous comment.

Line 79-81: “From 2015 to….”. Consider rephrasing, does not read well.

Line 81-83: I believe that some commas “,” are missing between “enforcement, further” and “actions, and…”.

Line 87: “Ouréns, Naya and Freire 2015” is a study carried out in Galicia (Spain) and seems not to be well referenced in this context, because you mention “…in this area”. Perhaps adding an “e.g.” when citing Ouréns et al 2015 is enough to solve this issue.

Line 89: “…. of biology…” . Change to “… of the biology….”

Lines 91-96: It is not clear how the example of New Zealand management plans (with number of licences and fishing limitations ) relates to the previous paragraph were you mention that some sustainable sea urchin fisheries rely on a good overview of the biology and population dynamics. Consider further development of this paragraph, providing the reader info on how scientific knowledge was incorporated into the management strategies there.

Line 94: “…fishing gears etc.)”. Add “,” between “fishing gears” and “etc.”

Line 129-139. Consider reviewing this paragraph as it is not easy to follow. Include references for the monitoring studies mentioned in lines 129-131. Authors refer to Pieraccini et al 2016 to state that the period between 2004 and 2007 can be considered the “pre-crisis era”. Yet, Pieraccini et al 2016 suggest that sea urchin stock contraction occurred between 2004-2005. As so, the period 2004-2007 encompasses that contraction in bigger individuals, and might not represent a reference environmental status as authors suggest in this manuscript.

Line 140-142. It is not clear the pertinence of this statement. The authors should consider rewriting this paragraph and clearly state the main goals and hypotheses being tested in the present study.

Material & Methods

Lines 147-153. Authors refer to Fig.2 and Fig. S1. Fig2 does not provide a general view of the MPA limits making it hard to interpret the spatial distribution of the study area. In my opinion Fig.2 should merge both figures (e.g. using fig s1 as the inset in fig2), so that the reader could follow this paragraph by looking at just one figure.

Line 154-162. This paragraph provides relevant background information on the sea urchin fishery regulations that could have been included in the introduction section for contextualization of the problem. Here, the intention seems to be a description of the harvesting pressure variability along the study area (inside/outside the MPA), and could be made more explicit and using comparable units of maximum allowed harvesting effort for ease of interpretation.

Line 164. “… (Fig.S1)…”. Why refer to a supplement figure? See previous comment on merging Fig.2 and Fig.S1.

Line 180. Typo. Remove “,”.

Line 189-196. This paragraph provides relevant background information on the sea urchin fishery regulations that could have been included in the introduction section for contextualization of the problem.

Line 200-201. “… size class ranges…”. For ease of interpretation, please provide numerical ranges for each size class considered.

Line 219. “… how are structured.” Change to “… how are they structured.”

Line 220. Typo. “suiTable”

Line 222-223. “These types of habitat…”. Does not read well.

Line 231-246. Why didn’t you run the model for the period 2004-2007 which is the period of interest in the present study? You mention that you only consider data between January and June, because it corresponds to the period active recruitment. Why this time frame and what do you mean by active recruitment? Don’t you mean settlement? In lines 264-265 you define recruits as individuals <2cm that survived ~2 years after settlement. It is not clear to me why the characterization of average bottom water speed only take into account this six months period.

Line 251-254. “In the pre-crisis…”. This should be described in the results section.

Line 280. “All the analysis…”. Change to “analyses”.

Line 284. “…no-parametric…”. Change to “non-parametric”.

Results

Line 306-317. Consider rephrasing. It is not easy to follow the description of these results.

Line 320-325. No easy to follow. Consider showing results in a graph instead of referring to supplement tables.

Lines 327. “Table S3”. Change to S2.

Line 332. “Fig. 4…”. Check unit labels in y axis of fig.4.

Line 331-336. Not easy to follow these results by looking at figure 4 and table S4. Table S4 lacks title and legend.

Line 348-351. “…General Linear Model…”. I would expect to see some model selection procedure that would eventually lead to a more parsimonious GLM model being fitted to the data without the non-significant explanatory variables being included in the final model. Can you please comment on this.

Line 350. “…density distribution of on the future stock….”. It is not clear what authors mean.

Discussion

Line 359-360. “…CR-1 was 1.5 and 1.6 times higher than…”. But according to the results not significantly different.

Line 367. “Surprisingly…”. Why surprising? Please expand on this discussion.

Line 374. “… when recruitment…” Do you mean settlement? Please make sure the concepts are coherent with what is described in the manuscript. In this study you don’t provide any data on settlement intensity, and the relationship established between recruitment (ind. <2 year old) and current speed may be the result of variability in both pre and post-settlement processes.

Line 384-385. “…1.2 higher than in sector 1…”. Are these differences significant?

Line 406-410. It is not clear to me what is meant in this paragraph. Consider rephrasing.

Line 412.”…then…”. Change to “in”.

Line 419. “…. Population dynamic…” . Change to “dynamics”

Line 423.”… …shed new light that might help management plans…”. How? Please develop.

Line 443-446. Doesn’t read well. Consider rephrasing

Conclusion.

Authors should also consider mentioning the importance of sound fishery related data (besides ecological studies) and active management tools that could allow to timely responses/adjustments to harvesting pressure in the region in order to promote sustainability of the resource and of this fishery.

References

Please thoroughly check references as there are several typos and inconsistencies in the format.

Figures and tables (main text and supplement)

Please thoroughly check all titles, legends, axis labels and units. All suplement material lack descriptions.

Raw data and code

Please check all raw data provided. It is a hard dataset to deal with. Should be curated before submitting.

Reviewer 2 ·

Basic reporting

The manuscript in the present form is of difficult reading, mostly due to the English language, that should be improved in order to allow a better flow of the manuscript and to ensure that the audience can clearly understand the text. Not clear and unambiguous, professional English has been used throughout. The litterature references are sufficient, some suggestion of additional references is provided to authors.

Experimental design

The main aim of the study, providing an ecological based approach for the management of P. lividus commercial harvesting, is a particularly interesting and current topic. Additionally, the amount of data gathered and variables considered are definitely relevant.
In my view, several clarifications in the different section of the manuscript are needed in order to clarify the different steps of the work done, the rationale behind some analyses and better explain results obtained. Some suggestion about clarifications needed are provided.

Validity of the findings

I think that authors should make more explicit which are the specific indications for fishery management plans provided by the present study. The conclusions should be more focused on main findings and possible use of the evidences from this study in management plans.

Additional comments

ABSTRACT: The abstract section in general is very confused and the aim of the study is not very clear from it. I think it should be entirely revised, in view also of following comments on the article sections. For example, the very first sentence (line 16) is very general… add a geographical reference (in the Mediterranean? Worldwide?). Lines 16-17: Which “crisis” are authors talking about? Make it clear for the readers. Line 22: replace “the environment” with “environmental factors/features”
INTRODUCTION: Line 40: add reference about “fish-sea urchins-macroalgae” trophic interaction. Line 44: remove “ecosystem” (barrens is sufficient). Line 54: this is valid also in the Mediterranean Sea? Please add references. Line 63: I think it could be useful mentioning that P. livisud is an “edible” species (instead of functional, given that its ecological role is explained in the following sentence). Line 70: “P. lividus” instead of this species. Lines 79-81: it is not clear the period of time to which authors are referring here. Lines 81-83: this sentence is not clear at all. Line 83: “preservation” instead of preserve. Lines 91-93: get rid of the first “management system” after “quota”. Line 102: “varies” instead of “vary” Line 134: pre-crisis era is a term present in literature or created by the authors? Probably it could be useful to explain better the concept and clarify why it should be used as a reference period. Lines 140-142: authors have not yet presented results, so it is not probably the case of concluding the introduction mentioning the “not negligible results”. Instead, it could be useful to state an aim of the study, e.g. “We used pre-crisis data to provide a reference framework to be used for planning future management strategies…”
MATERIALS & METHODS: Line 151: provide information about the year of establishment of the MPA. Line 154: add the temporal information about the pre-crisis era. Line 157: “the” before years should be removed. Lines 158-160: the regional decree is dated 2009, but the studied period is 2004-2007. The regulation outside the MPA was the same also before 2007? Line 199: add the information that the “types of habitat” will be described later in the following paragraph. Lines 200-201: define the size class ranges. Lines 204-207: this sentence is very difficult to follow, pease rewrite improving clarity. Additionally, explain the rationale behind the comparison between bottom current speed and recruit density and between predatory fish density and middle-size urchin density. Lines 240-242: explain why 2009-2010 water circulation data are appropriate for the 2004-2207 period. Lines 264—269: why the size of urchins is provided with letters and not with the actual numbers? It would be clearer. Additionally, I suggest to present data in fig. 4 presenting data grouped per size classes, or at least add this information with rectangles or different colors in the graphs. Lines 271-272: the anova on whole sea urchins’ density has been performed separately for the two factors “sector” and “habitat”? Lines 272-274: why analyses on the different size classes has not been performed considering the same factors? Please define “population”. Lines 284-287: as before, please explain the rationale behind these comparisons. Lines 288-289: Please describe the model: response variable, predictors….
RESULTS: Lines 343-345: the correlation is significant given the p-value, but it is very low (Spearman coeff. = -0.3). Lines 348-351: the model results should be described more in details. Assessment of goodness of fit of the model should be reported (e.g. % of variance explained; adjusted R2).
DISCUSSION: Lines 358-360: in the discussion session in my opinion authors should be more explicit and avoid acronyms in order to improve and facilitate the understanding of the readers. Here authors should for example refers to differences among the same type of habitat in the different sectors, with higher densities in calcareous rock in Sector one compared to the same habitat in sector 2 and 3. Lines 365-367: results of the model should be better explained. It has been presented as a predicting model: which predictions are derived from it? Line 376: the correlation is very weak. Line 400: how far larvae can spread? How much of the studied area is affected by larval dispersion? Lines 422-423: which are the indications for management provided by the present study?
CONCLUSIONS: I think that conclusions should be more focused on main findings and possible use of the evidences from this study in management plans. “Strong natural relationships” are mentioned: to which relationships authors are referring to?

---

## Round 0.2 · Minor Revisions

I have heard back from the more critical referee who feels that the manuscript is greatly improved from the initial submission. However, they still point out a number of issues with the resubmission that ought to be addressed prior to publication, and in particular, recommend that the manuscript be edited by a native speaker to improve the flow for the English usage and grammar. I expect that another round of careful revision should bring the manuscript to the level expected for publication, and I look forward to seeing your revised manuscript.

Reviewer 2 ·

Basic reporting

The present version of the manuscript is definitely improved compared to the first submission. Authors clarifiied some ambiguous aspects and explicited hypotheses.
Literature references are properly reported. The article structure is in line with PeerJ requirements and figures and tables are correctly provided. Raw data are shared.
Introduction and Discussion sections are quite long, partially because of some repetition and not perfect organization of the narrative buildup.
I still think that the manuscript would greatly benefit from a proofreading by a native speaker, in order to better organise the different sentences/sections, have a better flow and make it more easily readable.

Experimental design

The main aim of the study, providing an ecological based approach for the management of P. lividus commercial harvesting, is a relevant and current topic.
Research questions are in the present version better defined and several clarification about the experimental design were provided, improving the manuscript.
Rigorous statistical analyses have been performed and methos are described in full details.

Validity of the findings

Results are now better discussed in the management perspective, which is the aim of the study. Relevant insight about management recommendations are provided in the discussion.

Additional comments

The manuscript in the present version is definitely improved, but sometimes is still hard to read, in particular concerning Introduction and Discussion. I suggest to the authors to re-read the manuscript trying to avoid repetition and redundancy. A proofreading by a native speaker would be a great help in further improving the manuscript, which is relevant and plenty of useful information, sometimes hard to understand/catch because of the writing (I had to re-read some sentences several times to get the correct meaning).
Some specific comments on the text:
ABSTRACT:
Line 20: get rid of "On this island of... ". It is sufficient "In Sardinia..."
Line 33: get rid of "Also" (in general, do not use also at the beginning of a sentence!)
Lines 34-35: change "An abundance of different size classes..." with "Size-frequency distribution of sea urchins..."
Posidonia should be in italics

INTRODUCTION:
Line 61: after "target species" add examples: "target species SUCH AS...."
Line 65: I would change "However" with "On the other hand..."
Line 82-83: I would report as "area-specific"
Line 95: get rid of "locally" or move it before "considered"
Lines 95-98: merge the to sentences in order to make it clear that the regional decree refers to Sardinia; you may simply add "where" before "populations" in line 95, and change the full point with a comma in line 97.
Lines 101-102: I would change and simplify the sentence as "More restrictive regulation were enforced within marine protected areas (e.g....)
Line 106: get rid of "of it"
Line 173: "from when they were still undamaged" it is not clear to me, please explain better
Lines 194-195: but even larger size were sampled, even if not studied in relation to environmental factors. I suggest to clarify this point here and in M&M section
Line 196: change "in according to" with "for each"

MATERIAL & METHODS
Lines 242-245: move at the beginning of the following paragraph "Environmental constraints"
Lines 260-265: define better patches of what...
Lines 281-284: Move this sentence before the previous one (after the sentence ending with Marra et al 2016, line 279)
Lines 298-299: "...commercial stock and this size-class range is not considered in the analysis..." please explain better how you decided to use data, because those data about commercial size urchins are reposted as frequency distribution in describing population structure, and are correctly reported also in the result (are relevant data at is a pity to state that tey have been neglected, when they actually were not).
Lines 321-327: I am a little bit worried about auto-correlation of predictor variables. maybe a thouth about it should be useful

RESULTS:
Line 343: change "was" with "were"
Paragraph "Sea urchin population structure": data about the commercial size class (>5cm) are reported (and they are relevant!): please correct the description in M&M

DISCUSSION:
The discussion is improved from the previous version, but is very repetitive. I suggest to shorten it.
Line 399-400: change "OF Posidonia oceanica" with "CHARACTERISED BY Posidonia oceanica"
Line 404-405: get rid of "the high values of patch density and mean patch size seem to further increase the..." and change the sentence as "larger and more dense patches seems to further enhance sea urchin density...". Get rid of "sector 1" at the end of the sentence.
Line 406: start the sentence with "In fact, ..."
Lines 409- 414: these sentences are very repetitive and difficult to follow: try to summarise, merge sentences and expain better the concepts
Line 488: "their" instead of "its"
Line 495: some thing is missing between "unknown" and "the period"
Line 510: I would change "information and data" with "knowlwdge"

---

## Round 0.3 · accepted · Accept

I have read through your response referee feedback and your revised manuscript and believe that you have addressed all the referee concerns.
Therefore, I am happy to accept your paper and move it forward into production.